# Comprehensive Genomics Investigation of Neboviruses Reveals Distinct Codon Usage Patterns and Host Specificity

**DOI:** 10.3390/microorganisms12040696

**Published:** 2024-03-29

**Authors:** Rahul Kaushik, Naveen Kumar, Pragya Yadav, Shubhankar Sircar, Anita Shete-Aich, Ankur Singh, Shailly Tomar, Thomas Launey, Yashpal Singh Malik

**Affiliations:** 1Biotechnology Research Center, Technology Innovation Institute, Masdar City, Abu Dhabi P.O. Box 9639, United Arab Emirates; thomas.launey@tii.ae; 2Diagnostics and Vaccines Group, ICAR—National Institute of High Security Animal Diseases, Bhopal 462021, Madhya Pradesh, India; naveen.kumar4@icar.gov.in; 3Maximum Containment Facility, ICMR—National Institute of Virology, Pune 411001, Maharashtra, India; hellopragya22@gmail.com (P.Y.); sheteaich.a-niv@gov.in (A.S.-A.); 4Department of Animal Sciences, Washington State University, Pullman, WA 99163, USA; shubhankar.sircar@wsu.edu; 5Department of Biosciences and Bioengineering, Indian Institute of Technology Roorkee, Roorkee 247667, Uttarakhand, India; ankursingh.aec@gmail.com (A.S.); shailly.tomar@bt.iitr.ac.in (S.T.); 6College of Animal Biotechnology, Guru Angad Dev Veterinary and Animal Science University, Ludhiana 141004, Punjab, India

**Keywords:** *Neboviruses*, host specificity, codon usage patterns, genomic characterization, emerging viruses

## Abstract

*Neboviruses* (NeVs) from the *Caliciviridae* family have been linked to enteric diseases in bovines and have been detected worldwide. As viruses rely entirely on the cellular machinery of the host for replication, their ability to thrive in a specific host is greatly impacted by the specific codon usage preferences. Here, we systematically analyzed the codon usage bias in NeVs to explore the genetic and evolutionary patterns. Relative Synonymous Codon Usage and Effective Number of Codon analyses indicated a marginally lower codon usage bias in NeVs, predominantly influenced by the nucleotide compositional constraints. Nonetheless, NeVs showed a higher codon usage bias for codons containing G/C at the third codon position. The neutrality plot analysis revealed natural selection as the primary factor that shaped the codon usage bias in both the VP1 (82%) and VP2 (57%) genes of NeVs. Furthermore, the NeVs showed a highly comparable codon usage pattern to bovines, as reflected through Codon Adaptation Index and Relative Codon Deoptimization Index analyses. Notably, yak NeVs showed considerably different nucleotide compositional constraints and mutational pressure compared to bovine NeVs, which appear to be predominantly host-driven. This study sheds light on the genetic mechanism driving NeVs’ adaptability, evolution, and fitness to their host species.

## 1. Introduction

In the intricate world of virology, understanding the genetic- and protein-level makeup of viruses is crucial for unraveling their evolutionary patterns, host adaptation behavior, transmission potential, and impact on animal as well as human health [1,2,3]. Among the diverse array of enteric viruses that affect animal populations, Caliciviruses emerge as intriguing viruses that have piqued the interest of scientists and veterinarians alike [4,5,6,7,8]. With their potential to cause gastrointestinal disturbances in animals, Caliciviruses have surfaced as a significant concern for both animal health and the agriculture industry [9,10,11,12,13]. 

The members of the *Caliciviridae* family are highly diverse in genetic makeup and the family has been classified into eleven genera, viz. *Lagovirus*, *Vesivirus*, *Nebovirus*, *Sapovirus*, *Norovirus*, *Bavovirus*, *Minovirus*, *Nacovirus*, *Recovirus*, *Salovirus*, and *Valivirus* [10,12,13,14,15]. The members of the family are responsible for causing a spectrum of diseases in various animals and humans, such as respiratory diseases, hemorrhagic diseases, and gastroenteritis [16,17,18,19,20,21]. Nonetheless, among the enteric viruses, *Neboviruses* (NeVs), which primarily cause calf diarrhea, have created more interest among the research community because of their huge genetic diversity and widespread geographical distribution. Till now, NeVs have been identified in more than 14 countries [4,9,14,15,22,23,24,25,26,27]. Furthermore, our previous study confirmed the presence of NeVs in the Indian bovine population and whole-genome characterization revealed a highly divergent NeV strain [27]. Despite the huge genetic diversity and widespread geographical distribution of NeVs, a detailed dissection of factors governing the NeVs’ evolution, particularly host-driven evolutionary pressures, as well as their codon usage preferences, has not been performed so far. 

Codon optimization is a molecular biology technique used to enhance the expression of a gene in a particular host organism by altering the nucleotide sequence of the gene without changing the encoded amino acid sequence [2,3,8]. Viruses rely on the host cell’s machinery to replicate and produce viral proteins. Codon optimization aims to adapt viral genes to the codon usage patterns of the host organism, thereby facilitating efficient translation and protein production. Codon optimization involves replacing rare codons in the viral genome with synonymous codons that are more frequently used by the host organism to ensure efficient translation of the viral RNA in the host cell [2,5,8,20]. By adapting the viral genome to the host’s codon usage preferences, codon optimization can significantly increase the expression levels of viral proteins within host cells. Further, viruses with codon-optimized genomes are better equipped to exploit host cellular machinery for replication and propagation, enhancing fitness for more efficient viral spread and pathogenesis. Codon optimization can facilitate the adaptation of viruses to new host species. Overall, codon optimization is a powerful tool in viral research, enabling scientists to tailor viral genomes for efficient replication in specific host organisms, thereby advancing our understanding of viral pathogenesis and aiding the development of vaccines and antiviral therapies [2,8,16,20,22].

In fact, viruses undergo evolutionary pressures driven by their dependence on the host’s cellular machinery for survival. Consequently, they exhibit evolutionary signatures reflective of the host’s microcellular environment [3,28,29]. These distinctive evolutionary imprints contribute to the genetic and biochemical adaptation of viruses, enabling them to effectively circulate within their hosts, particularly in clinical and reservoir hosts [29]. Moreover, these viral evolutionary signatures have been shown to impact various biological processes in their hosts [2,3,30]. Investigating and understanding these signatures is crucial for gaining insights into the evolutionary mechanisms of synonymous codon usage and host-adapted evolution.

Therefore, in this study, we employed a panel of genetic tools, such as the Effective Number of Codons (ENC), neutrality and PR2-bias plots, Relative Codon Deoptimization Index (RCDI), Relative Synonymous Codon Usage (RSCU), and Codon Adaptation Index (CAI) to systematically analyze evolutionary signatures governing the host adaptation in highly divergent NeVs. We also looked for these signatures across the different proteins encoded by NeVs.

## 2. Materials and Methods

### 2.1. Whole-Genome NeV Dataset and Phylogenetic Analysis 

The complete genomes of all the NeVs (*n* = 16) available in the National Center for Biotechnology Information (NCBI) were downloaded, the list of which is summarized in Table 1. Additional metadata related to the assembly of the complete genomes accounted for in Table 1 are provided in Appendix A. The representative sequences for each of the eleven genera within the *Caliciviridae* family were also extracted from the NCBI and then aligned using MAFFT v.7.475. [31]. The LG-Gamma substitution model, incorporating empirical amino acid frequencies (LG + G + F), was identified as the most suitable amino acid substitution model for the dataset, determined by the Bayesian Information Criterion (BIC). Subsequently, phylogenetic trees were constructed using Maximum Likelihood (ML) inference and ultrafast bootstrap with 1000 replicates, employing IQ tree v2.1.2 [32]. The multiple sequence alignment of the VP1 and VP2 protein sequences was further used for identifying conserved and non-conserved regions. The nucleotide sequences corresponding to conserved and non-conserved amino acid residues were extracted from the genomic sequences of the different NeVs. These nucleotide sequences were used for performing codon usage bias in conserved and non-conserved regions in NeVs.

### 2.2. Nucleotide Composition Analysis

The VP1 and VP2 genes of NeVs were analyzed in terms of their nucleotide composition, total GC content, GC content at the first, second, and third codon positions (G1, G2, G3), and third synonymous codon position (T3s, C3s, A3s, and G3s) using the standalone version of codonW [33] and the utilities in EMBOSS [34]. The potential compositional bias in the VP1 and VP2 genes of NeVs was investigated and compared between other NeVs.

### 2.3. Effective Number of Codons (ENC) and ENC-GC3 Plot Analysis

The Effective Number of Codons (ENC) is a simplistic estimation of codon usage that varies from 20 to 61. An ENC value of 20 suggests the usage of only single codons for each amino acid, while a value of 61 indicates the equal contribution of all synonymous codons in coding corresponding amino acids [35,36,37]. A lower value of ENC (i.e., ENC < 40) specifies a strong codon usage bias in viruses [3,28]. 

A comparison of observed (oENC) and expected (eENC) ENC distribution at different GC3 values helps in estimating the role of selection pressure in shaping codon usage [35,36,37]. The expected ENC (eENC) was calculated using Equation (1).
(1)$$eENC=2+S+\frac{29}{\left(2S^{2}-2S+1\right)}$$

The values for *eENC* were calculated at different GC3 contents (*S*), and the values can vary from 0 to 1. The observed ENC (oENC) was calculated by implementing the standalone version of codonW [33] with the help of simple shell scripts. 

### 2.4. Neutrality Plot (Neutral Evolution) and PR2-Bias Plot 

In absence of any external pressure, mutations at the first, second, or third codon positions are expected to be equally likely. However, mutations at the first and second codon positions may lead to changes in amino acids (non-synonymous mutations), while third-codon-position mutations mostly result in the same amino acids (synonymous mutations). The neutrality plot analysis was implemented to examine the mutation–selection equilibrium in determining codon usage bias [38,39,40,41,42,43,44]. The average GC content at the first and second positions (GC12) was plotted against the average GC content at the third codon position (GC3) to compare the impacts of natural selection and mutation pressure on codon usage of protein-coding sequences. The required values for neutrality plot analysis were either already calculated in Section 2.3 or derived from these values using simple shell scripts. The values of GC12 and GC3 for the VP1 and VP2 genes of different NeVs were plotted to approximate the neutral evolution of the genes, where the slope of the regression line represents the magnitude of mutation pressure (slope = 0 suggests no effect of mutational pressure) and natural selection (slope = 1 suggests complete neutrality), while the regression coefficient against GC3 represents the mutation–selection equilibrium coefficient [45,46,47,48].

It has been demonstrated that the AT and GC appeared in pairs at the third codon positions under the influence of natural selection in codon usage [43,48]. To investigate this further, the GC bias (G3/(G3 + C3)) and AT bias (A3/(A3 + T3)) were calculated for the VP1 and VP2 genes of the NeVs. The values for G3, C3, A3, and T3 were calculated in Section 2.3. An unequal distribution of GC and AT compositions in protein-coding genes suggests the role of mutation pressure in codon usage, which can be further analyzed by scrutinizing the directionality of GC and AT bias [39,42,49,50,51,52]. Therefore, we employed Parity Rule 2 (PR2)-bias plots to investigate the GC and AT bias in the VP1 and VP2 genes of NeVs. 

### 2.5. Relative Synonymous Codon Usage (RSCU) 

The estimation of synonymous codon usage for each codon is investigated with the help of Relative Synonymous Codon Usage (RSCU) analysis. The RSCU values for a codon equaling unity (RSCU = 1) indicate an absence of any codon usage bias, while the higher values (RSCU > 1) suggest the presence of positive bias (preferred codon), and lower values (RSCU < 1) suggest the presence of negative bias (non-preferred codon) [45,49,53,54,55,56,57]. The RSCU values for 59 codons were calculated with the help of CAIcal [58] and codonW [33]. 

### 2.6. Codon Adaptation Index (CAI)

The Codon Adaptation Index (CAI) helps in the estimation of the relative adaptation of a viral gene to the codon usage of its host. The CAI values vary from 0 to 1, where a higher value suggests the usage of the most preferred codons among the host and virus. It is a generally agreed parameter for quantifying the degree of resemblances between codon usage of a gene and a reference dataset to investigate synonymous codon usage. Also, an estimate of gene expressivity, an examination of the factors governing synonymous codon usage, and an investigation of horizontal gene transfer have been performed with the assistance of CAI values [53,58,59,60,61,62]. Therefore, we employed CAI to investigate the codon usage preferences of NeVs in relation to seven host species, viz. *Bos taurus*, *Bubalus bubalis*, *Capra hircus*, *Ovis aries*, *Equus caballus*, *Equus asinus*, and *Camelus dromedarius*. The codon usage of the hosts was extracted from the Codon Usage Database [63], whereas the codon usage of NeVs was calculated by codonW [33] and CAIcal [58]. The frequencies of tRNAs in different hosts were retrieved from the GtRNAdb database (http://gtrnadb.ucsc.edu, accessed on 22 March 2024). Notably, the tRNA frequencies were available for only three hosts (out of the 7 different hosts considered in this study), viz. *Bos taurus*, *Equus caballus*, and *Ovis aries*.

### 2.7. Relative Codon Deoptimization Index (RCDI) 

The RCDI provides an inference to virus–host phylogenetic relationships and helps in deducing the possible host range of a virus. Also, a high RCDI reflects the possibility of expression of a gene in dormant stages and the presence of virus at a low replication rate in the hosts [5,46,49,64]. The ratio of observed RCDI (oRCDI) to the eRCDI offers a degree of virus–host adaptability, where a ratio close to 1 or higher represents better adaptability as compared to lower values. The synergistic effect of codon usage on gene expression and the plausible coevolution of the virus and its host genomes may also be indicated with the help of the RCDI of different genes for different hosts [2,5,28,29,44,46,57,64]. 

The Relative Codon Deoptimization Index (RCDI) values of the VP1 and VP2 genes of NeVs were calculated for an approximation of their similarity in codon frequencies against the codon frequencies of different hosts. The expected RCDI (eRCDI) was derived by using randomly generated sequences with the same G + C and amino acid compositions as the VP1 and VP2 genes in different NeVs. The observed RCDI (oRCDI) was directly calculated from the VP1 and VP2 gene sequences. The oRCDI and eRCDI were calculated by using the RCDI/eRCDI web server, available at http://genomes.urv.cat/CAIcal/RCDI, accessed on 22 March 2024 [64]. 

### 2.8. Recombination Analysis of NeVs

To investigate the recombination events among different strains of NeVs, RDP (version 4.1) was used [65,66]. The detection and characterization of recombination events in RDP (version 4.1) were accomplished by a simultaneous implementation of a diverse range of recombination detection methods that could deliver the desired efficiency despite the absence of a user-defined set of non-recombinant reference sequences. The multiple sequence alignment of all NeVs’ whole-genome sequences was performed by using MUSCLE [67] and this alignment was used as an input for recombination analysis in RDP (version 4.1) by selecting seven available methods for recombination detection, i.e., RDP [65], GeneConv [18,68], Chimera [69], MaxChi [70], 3Seq [71], BootScan [72], and SiScan [73]. Since the available genome sequences for NeVs are restricted (*n* = 16), all the sequences were used for performing recombination analysis. The highest acceptable *p*-value setting differs for the different methods. As recommended by the developers of RDP, the highest acceptable *p*-value 0.05 was used with multiple comparison correction. Further, the boundaries of breakpoints were determined by using the BURT method in the immediate vicinity of the identified recombination events along with implementing internal and external references.

## 3. Results

### 3.1. Nucleotide Composition and Codon Usage Bias Are Significantly Different in NeVs Infecting Yaks than Those of Bovines

The compositions of different nucleotides at the third synonymous codon position in NeVs are shown in Figure 1 for VP1 and VP2 genes. The composition analysis of the VP1 and VP2 genes of different NeVs indicated a GC-rich nature of the VP1 and VP2 genes. Notably, all the codons of each open reading frame were analyzed in the present study. The VP1 gene of different NeVs showed a higher (0.74 ± 0.043) GC content at the third synonymous codon positions (G3s + C3s) as compared to the VP2 gene (0.66 ± 0.057). Notably, the NeVs infecting yaks carried a lower GC content for both VP1 (0.65 ± 0.004) and VP2 (0.53 ± 0.032) at the third synonymous codon positions as compared to the NeVs infecting bovines (VP1 = 0.76 ± 0.037, and VP2 = 0.68 ± 0.029). Furthermore, an inverse pattern was observed for the AT composition at the third synonymous codon positions (A3s + T3s) of bovine-infecting NeVs. However, NeVs infecting yaks were observed to be AT rich at the third synonymous codon positions (VP1 = 0.55 ± 0.004, and VP2 = 0.64 ± 0.031) as compared to NeVs infecting bovines (VP1 = 0.42 ± 0.027, and VP2 = 0.47 ± 0.032). Further, we analyzed the codon usage patterns for conserved and non-conserved regions of the VP1 and VP2 genes of NeVs. Appendix A summarizes the overall codon usage in conserved and non-conserved regions of the VP1 and VP2 genes of NeVs. Additionally, Appendix A provide codon usage in conserved and non-conserved regions of the VP1 and VP2 genes. It is observed that the conserved regions preferred 11 A/T-ending and 16 G/C-ending codons while non-conserved regions preferred 19 A/T- and 15 G/C-ending codons.

The ENC values for the VP1 gene of different NeVs ranged from 52 to 56, with an ENC average of 52.8 ± 1.51. However, the ENC values for the VP2 gene showed a more diverse range (47.5 to 57.5), averaging 53.7 (±2.71). The ENC values for both the VP1 and VP2 genes of bovine NeVs are lower (52.7 ± 1.870) than those of yak NeVs (56.6 ± 0.639). The observed ENC values for different NeVs are provided in Appendix A. The higher ENC values in yak NeVs suggest a negligible codon usage bias as compared to bovine NeVs. 

### 3.2. Both Mutational Pressure and Natural Selection Govern the Codon Usage Patterns Differently across the NeV Genes as Well as Host Species

Next, we estimated the impact of mutation pressure and natural selection on the codon usage bias in both VP1 and VP2 genes of NeVs using an ENC-GC3 plot. The observed ENCs (oENCs) and GC3s for VP1 and VP2 genes are plotted with the standard curve for eENCs (Figure 2A). In Figure 2A, the eENC curve denotes that the contribution of only mutation pressure dictates codon usage. The observed ENC values for both the VP1 and VP2 genes of NeVs are positioned away from the eENC standard curve, indicating the role of both mutational pressure and selection pressure in governing codon usage. Notably, the distribution of VP1 genes of NeVs is comparatively closer to the eENC standard curve than that of the VP2 gene, suggesting the differential magnitude of influence of mutational pressure on them [43,56]. Furthermore, the deviations of oENC from eENC at different values of GC3s (ΔENC = eENC_GC3s_ − oENC_GC3s_) of the VP1 and VP2 genes of NeVs are calculated to compare the role of mutational pressure in shaping codon usage (Appendix A).

A higher deviation is suggestive of less impact of mutational pressure than other factors and vice versa [29,35,36,37,45,49]. It is observed that the average deviation in eENC and oENC values is lower in the VP1 gene (4.0 ± 0.54) as compared to the VP2 gene (5.47 ± 2.44) of NeVs. However, it is important to note that this deviation is very high in the VP2 gene of BoNeV/YLA-2/2017/CHN/MH718886 (ΔENC = 11.2), BoNeV/M3641/2011/HUN/JX018212 (ΔENC = 9.1), and BoNeV/LZB-1/2017/CHN/MG599036 (ΔENC = 8.0), reflecting a more prominent role of natural selection. Additionally, both the VP1 and VP2 genes of yak NeVs clustered away from the bovine NeVs. A higher deviation is suggestive of less impact of mutational pressure than other factors and vice versa [29,35,36,37,45,49]. It is observed that the average deviation in eENC and oENC values is lower in the VP1 gene (4.0 ± 0.54) as compared to the VP2 gene (5.47 ± 2.44) of NeVs. However, it is important to note that this deviation is very high in the VP2 gene of BoNeV/YLA-2/2017/CHN/MH718886 (ΔENC = 11.2), BoNeV/M3641/2011/HUN/JX018212 (ΔENC = 9.1), and BoNeV/LZB-1/2017/CHN/MG599036 (ΔENC = 8.0), reflecting a more prominent role of natural selection. Additionally, both the VP1 and VP2 genes of yak NeVs clustered away from the bovine NeVs.

The neutrality plot analysis showed that the magnitude of mutational pressure In shaping the codon usage in VP1 genes of NeVs is 18% (R^2^ = 0.41, *p* < 0.01). However, in the case of VP2 genes of NeVs, a substantial contribution of mutational pressure (43%) was noted (R^2^ = 0.41, *p* < 0.01) (Figure 2B). It is also worth mentioning that the VP1 and VP2 genes of yak NeVs have considerably different compositions of GC12 and GC3 as compared to their counterparts isolated from bovines and clustered differently in the neutrality plot (Figure 2B). A narrow distribution of GC compositions reflected the dominance of natural selection over mutational pressure, while a significant correlation between GC12 and GC3 compositions signified the impact of mutational pressure at all the codon positions.

The role of mutation pressure in codon usage is further explored by scrutinizing the directionality of GC and AT bias through the distribution of GC and AT compositions in the VP1 and VP2 genes of NeVs [39,42,49,50,51,52]. The GC and AT biases in the VP2 genes of yak NeVs are quite different from those of bovine NeVs, which is in concordance with the neutrality plot (Figure 2C). The wide distribution of GC and AT bias in the VP1 and VP2 genes of NeVs suggests the differential mutational pressure’s contribution in shaping codon usage [47,52,74].

### 3.3. Most Common Codons across NeVs Have Abundant G/C at the Third Position

The RSCU analysis of NeVs showed that all the 13 most abundant codons common across NeVs have G/C at the third position (AAG, ACC, AUC, CAC, CCC, CGC, CUC, CUG, GAG, GCC, GGC, GUG, UCC). However, the most preferred codons among the bovine NeVs include an additional five codons (CAG, GAC, GGG, UAC, UUC). In the case of yak NeVs, the total number of preferred codons increased to 30, of which 12 codons have A/U at the third codon position. Overall, the RSCU analysis revealed that NeVs have a higher codon usage bias for the codons with G/C at the third codon position as compared to the A/U-ended codons. A correlation analysis of RSCU among the bovine NeVs indicated a highly similar codon usage pattern (average R^2^ = 0.75 ± 0.008, *p* < 0.001). However, the codon usage pattern in yak NeVs was found to deviate from the bovine NeVs (average R^2^ = 0.51 ± 0.003, *p* < 0.001). A heatmap representation of the RSCU analysis of 59 codons (excluding UGG, AUG, UAG, UAA, and UGA) of NeVs and their comparison with different host species are depicted in Figure 3. Although NeVs have been detected in bovines and yak species, other host species were also included in this analysis to understand their comparative codon usage similarity.

Furthermore, the correlation analysis between the RSCU of different NeVs and seven host species (*Bos taurus*, *Equus caballus*, *Equus asinus*, *Camelus dromedarius*, *Bubalus bubalis*, *Ovis aries*, *Capra hircus*) revealed that bovine NeVs have a higher preference for *Bos taurus*, indicating the practical utility of implementing RSCU analysis. A summary of the correlation analysis of RSCUs between different NeVs and potential hosts is provided in Appendix A and common preferred codons (out of 59 codons) in different strains of NeVs and different hosts in Appendix A.

The host specificity of NeVs is further explored in terms of the number of preferred codons that are shared by the host species and virus, where a higher number of preferred codons common in the host and virus suggests a better host adaptability [41,54,57,75,76,77,78]. For instance, the BoNeV/Mukti/2016/IND/MN241817 is observed to share 21 preferred codons with Bos taurus and Equus caballus. Likewise, for BoNev/Kirklareli/2012/TUR/NC_030793, 23 preferred codons were shared with *Bos taurus*.

### 3.4. Codon Adaptation Index Showed Marginal Variations among NeVs

The CAI of NeVs is the highest for *Bubalus bubalis* (0.71 ± 0.009 for VP1, 0.69 ± 0.027 for VP2), followed by *Bos taurus* (0.70 ± 0.009 for VP1, 0.68 ± 0.026 for VP2). However, the CAI is the lowest for *Camelus dromedarius* (0.65 ± 0.011 for VP1, 0.63 ± 0.024 for VP2). The overall CAI for NeVs is observed to be on the higher side and well suited to the bovines. However, the variations in the CAI among NeVs are found to be marginal, including the yak NeVs. The CAIs of different NeVs for VP1 and VP2 genes are provided in Appendix A, respectively. 

It may be inferred that the CAI values of the VP1 and VP2 genes of different NeVs for different hosts were observed to be high and did not deviate significantly. A controlled expression of viral proteins and replicative stability in different viruses is accomplished either by using optimized codon usage or by escaping the immune system of hosts [49,50,56,70]. In the case of NeVs, the higher values of CAI suggest the usage of optimal codons, indicating their replication stability in hosts. Among the different hosts, the VP1 and VP2 genes of *NeVs* reflect a slightly better adaptability to *Bos taurus*, and *Bubalus bubalis* as compared to other hosts.

### 3.5. Viral Adaptability to Different Hosts through RCDI 

Notably, the ratio of the observed RCDI to the expected RCDI is the highest for both the VP1 and VP2 genes of strain Mukti (accession id—MN241817) for *Bos taurus* (0.79 and 0.91), followed by *Equus callabus* (0.77 and 0.89). Nonetheless, the RCDI ratios for VP1 genes of other NeVs are also high for *Bos taurus* (0.78 ± 0.02) as compared to other hosts (*Capra hircus* 0.76 ± 0.06; *Equus callabus* 0.75 ± 0.02; *Bubalus bubalis* 0.74 ± 0.02; *Camelus dromedarius* 0.72 ± 0.02; *Ovis aries* 0.72 ± 0.01; *Equss asinus* 0.72 ± 0.01), as depicted in Figure 4. For VP2 genes, a similar trend of RCDI ratios was observed (Figure 4). The high RCDI ratios suggest a slightly better adaptability and specificity of NeVs to *Bos taurus* as compared to other hosts. The observed RCDI values and expected RCDI values for both the VP1 and VP2 genes of NeVs in relation to different host species are provided in Appendix A, respectively. 

### 3.6. Isoacceptor tRNA Pool Analysis of Different Hosts

Different codons that represent distinct amino acid residues are bound by distinct isoacceptor tRNAs of different tRNA species. Translation selection is determined by whether the majority of codons that NeVs prefer are recognized by the majority of isoacceptor tRNAs in the host. The most prevalent isoacceptor tRNAs identify the favored codons, demonstrating the impact of translational selection on codon usage. Understanding how neboviruses adapt to the tRNA pool of various hosts is crucial since translation is the primary mechanism in any virus life cycle. As a result, we examined the codons that the hosts’ tRNA pool liked [28]. When examining the synonymous codon families and the tRNA anticodon, it becomes apparent that, with the exception of Trp and Met, among nine two-fold synonymous codon families in NeVs, seven (Asn, Asp, Cys, Gln, Glu, His, Lys, Phe, and Tyr) exhibit optimal codon–anticodon usage (Appendix A). Specifically, the preferred codons within these families are associated with the most frequently occurring tRNA isotypes in bovine cells, indicating an optimal alignment of codon and anticodon pairs. In the rest of the synonymous codon families (Ala, Arg, Ile, Leu, Pro, Ser, Thr, Val), except Gly, NeVs exhibit non-optimal codon–anticodon usage. In contrast, two YAK NeVs carried distinct non-optimal codon–anticodon usage for Phe and Gln, and optimal for Ser and Val. These results highlight that NeVs have evolved to use combined codon–anticodon usage (optimal and non-optimal) of different host species’ tRNA pools.

### 3.7. Phylogenetic Analysis of NeVs Revealed Two Distinct Clusters

The genomic sequences may be used to deduce the evolutionary relationships among different organisms and these relationships may be represented in the form of phylogenetic trees. With the evolution and divergence of organisms, mutations are accumulated in the nucleotide sequences that may be compared via multiple sequence alignment to trace the evolutionary history [3,79,80,81,82]. The phylogenetic analysis of NeVs showed that NeVs formed two distinct clusters, where one cluster comprised Mukti, Kirklareli, and other NeVs recently detected in China in the year 2021, whereas the second cluster consisted of NeV strains from China, Japan, and Hungary, including the representative NeV strains (Figure 5). Additionally, to understand the identity and non-identity groups according to the amino acid translated, we created a pairwise homology plot for the VP1 and VP2 genes of neboviruses (Appendix A), which revealed a similar pattern of relationships to that indicated by phylogeny analysis.

### 3.8. Recombination Analysis Indicated Four Potential Recombination Events in NeVs 

RNA viruses are known to adapt to respective hosts in a more efficient manner as compared to DNA viruses due to their high mutation rates [83,84,85]. The favorable mutations are believed to occur because of errors in gene replication that enable the viruses to acclimatize to diverse environmental selection pressures. Recombination events generally integrate distantly related or unrelated sequences and pose a challenge for accurate reconstruction of the sequence-based phylogeny [7,84]. 

Four events of recombination were observed among the NeVs, where at least two out of the seven methods detected recombination events. For instance, only two recombination events involving ‘BoNeV/LZB-1/2017/CHN/MG599036’ and ‘BoNev/Newbury1/2005/UK/NC_007916’ were detected using all methods. An overview of detected recombination events, corresponding recombinants, major parents, minor parents, and methods showing positive and negative symbols for the presence or absence of recombination is provided in Table 2. Also, the genomic positions of major and minor parents for different recombination events are summarized in Appendix A. 

## 4. Discussion

The investigation of codon usage patterns in genomes has garnered significant attention due to its implications for understanding the molecular basis of gene expression, evolution, protein synthesis, and host adaptation [16,44]. Synonymous codon usage in different gene-coding regions is a consequence of codon usage bias and is governed through specific selection pressures such as natural selection, mutation pressure, and nucleotide compositional constraints [2,3,86]. Among these factors, the selective pressure from the host cells is opined to be very crucial in viruses [29,57]. The obligate intracellular parasitic nature of viruses subjugates them on the host cell mechanism for gene expression, and a comprehensive codon usage analysis of viral genomes may help in exploring the viral fitness and specificity to various primary and reservoir hosts [2,5,29,44,86]. However, the factors influencing the evolution of codon usage in NeVs, along with their preferences in codon usage in comparison with hosts, have not been explored to date. Therefore, in this study, we presented a comprehensive approach to systematically examine genome-wide codon usage patterns in NeVs. This examination extended to exploring these distinctive signatures in the various proteins encoded by NeVs.

To investigate the influence of nucleotide compositional constraints and mutational pressure on shaping synonymous codon usage bias, we extracted the complete genomes of 16 NeVs. It is important to acknowledge that biases or errors might be introduced by different sequencing technologies. Consequently, additional information about sequence depth could greatly enhance the interpretation of the results from assembled genomes. While the NCBI repository for complete genomes encourages submitters to provide such information as metadata, unfortunately, it is not available for the 16 NeVs, neither through metadata nor in the corresponding research articles. It is worth noting that some RNA viruses employ a strategy of population diversity to adapt to various host species, resulting in a spectrum of genotypes with differing frequencies across hosts. Hence, sequence depth emerges as a critical parameter for genetic analyses of viral genomic sequences. We performed an ENC plot analysis to investigate how the nucleotide compositional constraint and mutational pressure contribute to shaping the bias in synonymous codon usage within NeV genomes. This analysis revealed that the yak NeVs detected in China deviated significantly from other NeVs, indicating the profound influence of GC3s in compositional bias in yak NeVs. However, all the NeVs lay below the expected ENC curve, implying that mutation pressure is not the only primary factor governing the codon bias in NeVs; other factors also contribute to shaping the codon usage bias in NeVs. Therefore, we conducted a neutrality plot analysis to unravel the influence of crucial factors, namely natural selection and mutation pressure, on the bias observed in codon usage. Natural selection primarily dictated the codon usage shaping in both the VP1 (82%) and VP2 (57%) genes of NeVs. However, neutrality plot analysis also suggested that yak NeVs possess a considerably different composition of GC12 and GC3 as compared to their counterparts detected from bovines and cluster differently. This differential nucleotide compositional constraint and mutational pressure observed in yak NeVs seems to be primarily host-driven. The host-driven factors influencing the codon usage bias in viruses have been demonstrated in various previous studies [3,29,86].

The codon usage bias in multiple RNA viruses, such as Equine Influenza virus (ENC = 52.09) [28]; H5N1 Influenza virus (ENC = 50.91) [87]; Chikungunya Virus (ENC = 55.56) [88]; Nipah virus (ENC = 51.06) [89]; West Nile Virus (ENC = 53.81) [90]; and Foot-and-Mouth Disease Virus (ENC = 51.42) [91] showed that RNA viruses prefer lower codon bias, which is in concordance with our study (ENC = 52.7). The possible explanation of this lowered codon bias is that it provides an opportunity for RNA viruses to adapt to multiple hosts, whose microcellular environments offer completely different tRNA pools [28]. 

Furthermore, to elucidate the degree of preference for A/U- and G/C-ended codons and to assess the patterns of synonymous codon usage, we calculated the Relative Synonymous Codon Usage (RSCU) values for each codon across 16 strains of NeVs. These values were then compared among various host species. It is evident from RSCU analysis that NeVs possess a higher codon usage bias for the codons with G/C at the third codon position as compared to the A/U-ended codons. This contrasts with previous studies, where different RNA viruses prefer A- or U-ended codons [2,3,28,29,86,87,92]. To compare the preferences in codon usage between NeVs and various host species, we combine RSCU analysis with the Codon Adaptation Index (CAI) and Relative Codon Deoptimization Index (RCDI) analyses. These analyses clearly showed that NeVs possess codon usage patterns more similar to bovines.

Codon usage in viruses is also affected by the translation selection (tRNA abundance along with their modifications) in a particular host species [93], both of which can be quantified by nanopore RNA sequencing (nano tRNA-seq) [94]. Translational selection also influences the codon usage in viruses because the viruses show preferences for the codons that are identified by the most prevalent isoacceptor tRNAs in the host to ensure efficient translation of their proteins [28]. This way, viruses try to adapt to the tRNA pool of the host. Given that translation selection plays a pivotal role in the life cycle of any virus, it becomes crucial to understand how different NeVs adapt to the pool of different host species’ tRNAs. When examining the synonymous codon families and the tRNA anticodon, it becomes apparent that, with the exception of Trp and Met, among nine two-fold synonymous codon families in NeVs, seven (Asn, Asp, Cys, Gln, Glu, His, Lys, Phe, and Tyr) exhibit optimal codon–anticodon usage (Appendix A). Specifically, the preferred codons within these families are associated with the most frequently occurring tRNA isotypes in bovine cells, indicating an optimal alignment of codon and anticodon pairs. In the rest of the synonymous codon families (Ala, Arg, Ile, Leu, Pro, Ser, Thr, Val), except Gly, NeVs exhibit non-optimal codon–anticodon usage. In contrast, two yak NeVs carried distinct non-optimal codon–anticodon usage for Phe and Gln, and optimal for Ser and Val. These results highlight that NeVs have evolved to use combined codon–anticodon usage (optimal and non-optimal) of different host species’ tRNA pools.

Epigenetic modifications in viruses can influence codon usage by affecting the accessibility of viral genomes, regulating viral gene expression, and responding to host immune pressures [95]. Understanding these interactions is crucial for unraveling the molecular mechanisms underlying viral pathogenesis and evolution; however, studies linking specific epigenetic modifications to codon usage in viruses are limited.

This study convincingly illustrates that the intricate patterns of genome-wide codon usage in NeVs are shaped by a complex interplay of determinant factors, including mutation pressure, natural selection, nucleotide compositional constraints, and other unidentified factors. The codon usage bias in NeVs was found to be relatively weaker, primarily influenced by mutation pressure and nucleotide compositional constraints. Overall, the findings of this study contribute to a deeper understanding of the factors involved in shaping codon usage in NeVs and their fitness in relation to their host.

## Figures and Tables

**Figure 1 microorganisms-12-00696-f001:**
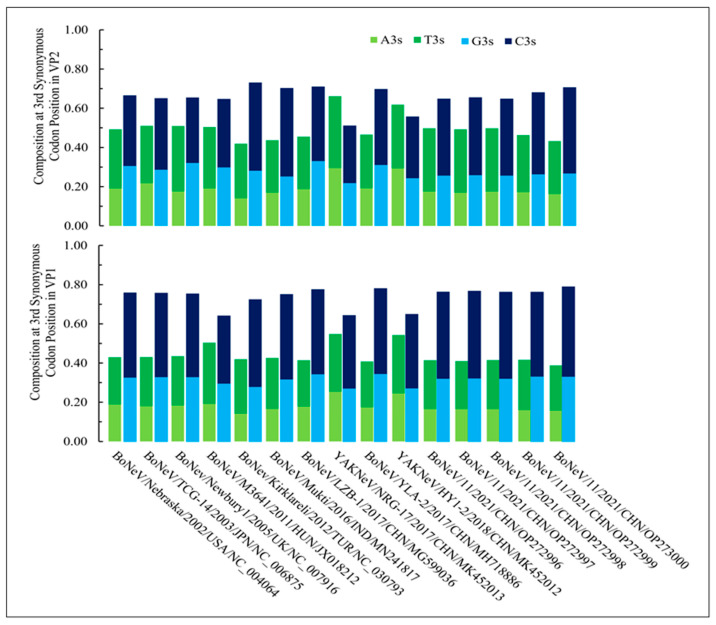
Composition analysis of VP1 and VP2 genes of different NeVs at the 3rd synonymous codon positions. Since the compositions are at the 3rd synonymous codon positions, these need not necessarily sum up to one.

**Figure 2 microorganisms-12-00696-f002:**
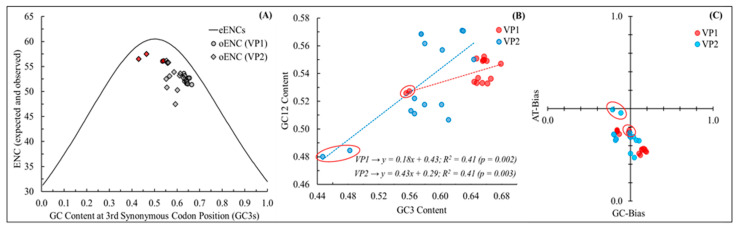
(**A**) ENC-GC3 plot for the VP1 and VP2 genes, where the circle- and diamond-shaped data points account for VP1 and VP2 genes, respectively. The grey and red colors represent the NeVs isolated from bovines and yaks, respectively. (**B**) Neutrality plot for the VP1 (red) and VP2 (blue) genes of different NeVs. The regression coefficient indicates an 18% and 43% effect of mutation pressure in shaping the codon usage in VP1 and VP2 genes of NeVs, respectively. (**C**) Parity Rule 2 (PR2)-bias plot for VP1 and VP2 genes of different NeVs, where deviation from equal GC and AT bias in the VP1 and VP2 genes suggests the contribution of mutational pressure in codon usage in NeVs. The values highlighted with red circles represent the NeVs isolated from yaks.

**Figure 3 microorganisms-12-00696-f003:**
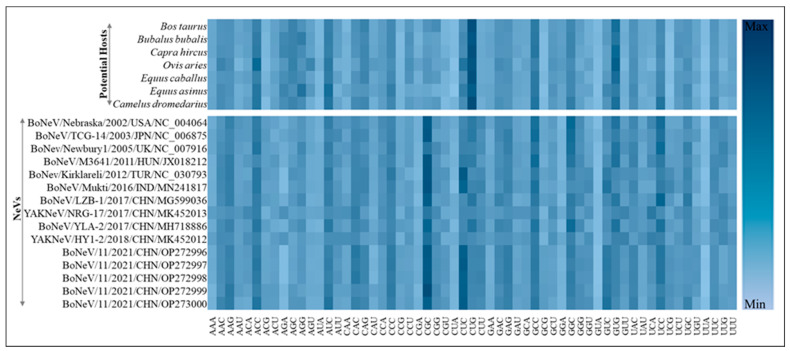
A heatmap representation of RSCU analysis of 59 codons (excluding UGG, AUG, UAG, UAA, and UGA) of NeVs and different host species.

**Figure 4 microorganisms-12-00696-f004:**
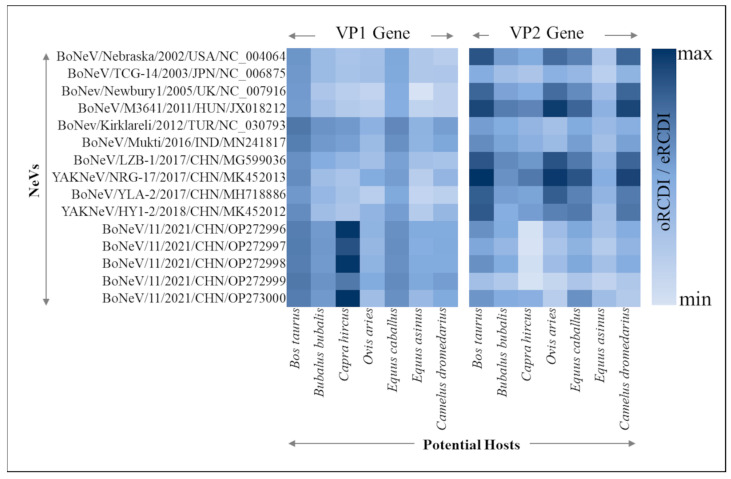
A heatmap depiction of ratios between observed and expected Relative Codon Deoptimization Index (oRCDI and eRCDI) of VP1 and VP2 genes of different NeVs with respect to different hosts, where the higher ratio indicates better viral adaptability to the corresponding host(s).

**Figure 5 microorganisms-12-00696-f005:**
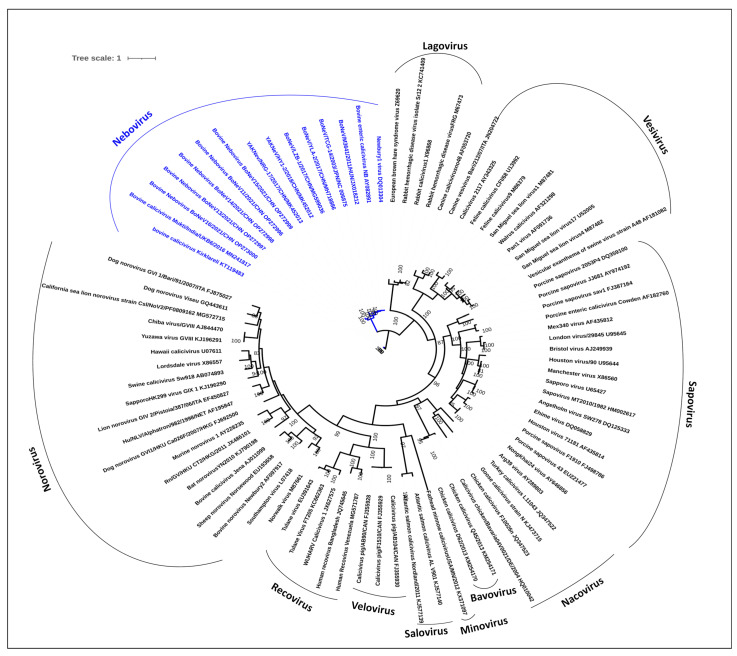
A phylogenetic tree depicting genetic relationships of NeVs within the *Caliciviridae* family.

**Table 1 microorganisms-12-00696-t001:** A summary of different strains of NeVs available on the NCBI resource portal as complete genomes. The GenBank identifiers, genome size in base pairs, start and end positions of ORF1 and ORF2 on the genome, and country of isolation of the viruses are provided. For the sequencing methods, NA represents the NeVs for which the information about method is neither available though public resources nor through corresponding research articles.

NeVs	Accession ID	Method	Size (bp)	Source (PMID)	Origin
BoNeV/Nebraska/2002/USA/NC_004064	NC_004064	NA	7453	12239283	USA
BoNeV/TCG-14/2003/JPN/NC_006875	NC_006875	NA	7453	NCBI	Japan
BoNev/Newbury1/2005/UK/NC_007916	NC_007916	NA	7454	16574184	UK
BoNeV/M3641/2011/HUN/JX018212	JX018212	Sanger	7453	NCBI	Hungary
BoNev/Kirklareli/2012/TUR/NC_030793	NC_030793	Sanger	7484	26292294	Turkey
BoNeV/Mukti/2016/IND/MN241817	MN241817	Sanger	7484	NCBI	India
BoNeV/LZB-1/2017/CHN/MG599036	MG599036	Sanger	7453	31500726	China
YAKNeV/NRG-17/2017/CHN/MK452013	MK452013	Sanger	7459	31500726	China
BoNeV/YLA-2/2017/CHN/MH718886	MH718886	Sanger	7453	30444471	China
YAKNeV/HY1-2/2018/CHN/MK452012	MK452012	Sanger	7460	31500726	China
BoNeV/11/2021/CHN/OP272996	OP272996	Illumina	7485	38108282	China
BoNeV/11/2021/CHN/OP272997	OP272997	Illumina	7453	38108282	China
BoNeV/11/2021/CHN/OP272998	OP272998	Illumina	7462	38108282	China
BoNeV/11/2021/CHN/OP272999	OP272999	Illumina	7466	38108282	China
BoNeV/11/2021/CHN/OP273000	OP273000	Illumina	7463	38108282	China

**Table 2 microorganisms-12-00696-t002:** A summary of recombination events observed from multiple sequence alignment of complete genome sequences of NeVs. The NeVs for which at least two of the selected methods (out of seven) detected the occurrence (symbolized with ‘+’) of recombination are shown here. The non-occurrence/detection is symbolized with ‘-’. The asterisk (*) in some of the major and minor parents indicates the sequence used to infer the unknown major or minor parent. The abbreviations for the labels in the first row are provided in the footnote to the table.

REs	Recomb	Major Parent	Minor Parent	RDP	GCn	BSc	MCh	Chm	SiS	3SQ
1	MG599036	JX018212	NC_007916 *	+	+	+	+	+	+	+
2	NC_007916	MH718886 *	NC_004064	+	+	+	+	+	+	+
3	NC_004064	NC_007916	MH718886	-	-	-	-	-	+	+
4	JX018212	MH718886	NC_004064	-	-	-	-	-	+	+

Abbreviations—REs: recombination events; Recom: recombinant; GCn: GeneConv; BSc: BootScan; MCh: MaxChi; Chm: chimera; SiS: SiScan; 3SQ: 3Seq.

## Data Availability

The original contributions presented in this study are included in the article and Appendix A; further inquiries can be directed to the corresponding authors.

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
