# Peer review of "Comprehensive Genomics Investigation of Neboviruses Reveals Distinct Codon Usage Patterns and Host Specificity"

_microorganisms, 2024, doi:10.3390/microorganisms12040696_

Round 1
Reviewer 1 Report
Comments and Suggestions for Authors
The manuscript "Comprehensive Genomics Investigation of Neboviruses Reveals Distinct Codon Usage Patterns and Host Specificity" presents an original bioinformatics analysis of virus sequence composition in relation to codon usage. The study's logic is sound, and the results offer valuable insights into neboviral evolution across different host species.
As a general recommendation, it is important to include the date of the last database and software consultation.
I have two main concerns:
11) In Table 1, it would be beneficial to include the methodology used to acquire the viral sequences (Sanger, NGS, etc.) and the depth of the reads, whether these data can be retrieved from sequence repositories or linked papers. Different sequencing technologies can introduce biases or errors, and understanding the sequence depth is crucial for interpreting the results accurately, because the assembled sequences could only represent an "individual" within a diverse population or "the most common consensus”. Without this information, additional analyses may be challenging. Besides, these considerations should be reflected in the presentation and discussion of results.
Some RNA viruses adopt a strategy of population diversity to survive in different species, resulting in a group of genotypes with varying frequencies depending on the host. Therefore, read depth becomes a critical parameter for genetic analyses of viral nucleic acids.
22) Another issue that would be interesting for the authors to reflect on in their work is differentiating the occurrence of variarions in codons between conserved and non-conserved amino acids in the proteins studied. Specifically, it is unclear to me whether only synonymous mutations were examined, focusing on conserved regions at the protein level in the polypeptides under study, or if all codons of each open reading frame were analyzed. I recommend providing greater clarity on this point. It is possible that some nucleotidic changes may be attributable to factors other than the availability of tRNAs (e.g. protein folding, protein partners, RNA folding, target sequences for miRNA, etc.). I believe that a protein homology plot, where codons are classified into identity and non-identity groups according to the amino acid translated, could offer valuable insights to detect possible codon usage biases in the first group.
Comments on the Quality of English LanguageThe manuscript is written correctly.
Author Response
Response to Reviewers’ Comments
Reviewer #1
The manuscript "Comprehensive Genomics Investigation of Neboviruses Reveals Distinct Codon Usage Patterns and Host Specificity" presents an original bioinformatics analysis of virus sequence composition in relation to codon usage. The study's logic is sound, and the results offer valuable insights into neboviral evolution across different host species.
11) In Table 1, it would be beneficial to include the methodology used to acquire the viral sequences (Sanger, NGS, etc.) and the depth of the reads, whether these data can be retrieved from sequence repositories or linked papers. Different sequencing technologies can introduce biases or errors, and understanding the sequence depth is crucial for interpreting the results accurately, because the assembled sequences could only represent an "individual" within a diverse population or "the most common consensus”. Without this information, additional analyses may be challenging. Besides, these considerations should be reflected in the presentation and discussion of results. Some RNA viruses adopt a strategy of population diversity to survive in different species, resulting in a group of genotypes with varying frequencies depending on the host. Therefore, read depth becomes a critical parameter for genetic analyses of viral nucleic acids.
Response: We are thankful to the reviewer for this suggestion. We have added methodology of sequencing, and source of data (reference to research article or public repository) in Table 1 of revised manuscript. However, the information about the read depth could not be retrieved from corresponding research article or public sequence repository. Further, the bias or errors imparted through different sequencing technologies and essence of the availability sequence depth is added to the discussion section of the revised manuscript (page 11, line 392-403).
22) Another issue that would be interesting for the authors to reflect on in their work is differentiating the occurrence of variations in codons between conserved and non-conserved amino acids in the proteins studied. Specifically, it is unclear to me whether only synonymous mutations were examined, focusing on conserved regions at the protein level in the polypeptides under study, or if all codons of each open reading frame were analyzed. I recommend providing greater clarity on this point. It is possible that some nucleotidic changes may be attributable to factors other than the availability of tRNAs (e.g. protein folding, protein partners, RNA folding, target sequences for miRNA, etc.). I believe that a protein homology plot, where codons are classified into identity and non-identity groups according to the amino acid translated, could offer valuable insights to detect possible codon usage biases in the first group.
Response: As per the reviewers’ suggestion, we have added the occurrence of variations in codons between conserved and non-conserved amino acids in the VP1 and VP2 proteins. Figure S1 in revised manuscript summarizes an overall codon usage in conserved and non-conserved regions of VP1 and VP2 genes of neboviruses. Additionally, Table S2 and S3 provides codon usage in conserved and non-conserved regions of VP1 and VP2 genes of different neboviruses. In the initially submitted version of the manuscript, all the codons of VP1 and VP2 were analyzed irrespective of nature (synonymous or non-synonymous) of mutations. The same is explicitly mentioned in the revised manuscript to avoid any ambiguity (page 5, lines 192-193, 202-208). Further, as per the suggestion, the protein homology plot for VP1 and VP2 proteins among the neboviruses is now provided as Supplementary Figure S2 in the revised manuscript (page 10, lines 351-354).
Reviewer 2 Report
Comments and Suggestions for Authors
Dear authors,
you presented a well-written manuscript about codon usage patterns and host specificity.
I have some minor revisions to improve the overall interest to the reader.
Please check for typos and double spaces (L78, 113, 190, 233)
Please include a paragraph in the introduction about the role of codon optimization, mechanisms and advantages for viral host adaption.
The materials and methods are adequate. However, is it possible to include tRNA quantification in these hosts (e.g. nano tRNA-seq) to further strenghten your results? If this is not possible please include a paragraph in the discussion. Also include here viral epigenomic modifications that are likely to occur due to the codon usage.
For the discussion, I am missing a clear Take-Home message. The fact that viral codon optimization occurs is clear for the reader. Point out your contribution
Comments on the Quality of English Language
See above
Author Response
Reviewer #2
Dear authors, you presented a well-written manuscript about codon usage patterns and host specificity. I have some minor revisions to improve the overall interest of the reader.
- Please check for typos and double spaces (L78, 113, 190, 233).
Response: We thank the reviewer for pointing this out. We have corrected the typos and double spaces in the revised manuscript.
- Please include a paragraph in the introduction about the role of codon optimization, mechanisms, and advantages for viral host adaptation.
Response: We are thankful to the reviewer for this suggestion. In the revised manuscript, we have included a paragraph in the Introduction section about the role of codon optimization, mechanisms and advantages for viral host adaptation (page 2, lines 60-73).
- The materials and methods are adequate. However, is it possible to include tRNA quantification in these hosts (e.g. nano tRNA-seq) to further strengthen your results? If this is not possible, please include a paragraph in the discussion. Also include here viral epigenomic modifications that are likely to occur due to the codon usage.
Response: We are thankful to the reviewer for this suggestion. We completely agree that the tRNA quantification in the potential hosts could strengthen the results significantly. However, non-availability of tRNA quantification data of these hosts in public repositories and limited experimental resources restricts the comprehensive analysis for all the hosts. Based on availability of tRNA pool, we have included Bos taurus, Ovis aries, and Equus caballus for correlating the codon usage bias with the most abundant isoacceptor tRNAs in these hosts (page 5, lines 174-177; page 11, lines 365-382; page 14, lines 482-507).
- For the discussion, I am missing a clear Take-Home message. The fact that viral codon optimization occurs is clear for the reader. Point out your contribution.
Response: We thank the reviewer for pointing this out. We have revised the discussion section of the manuscript to focus on the findings of the presented study.
Round 2
Reviewer 1 Report
Comments and Suggestions for Authors
The authors have taken into account all the observations I have expressed, contextualizing the results found with the availability of the analyzed genomic information. I have no further concerns to raise about the manuscript.
Comments on the Quality of English LanguageThe text in general is well written.